# Continuous Cropping Changes the Composition and Diversity of Bacterial Communities: A Meta-Analysis in Nine Different Fields with Different Plant Cultivation

**Mohammad Murtaza Alami** [1], **Qiuling Pang** [1], **Zedan Gong** [1], **Tewu Yang** [1], **Daiqun Tu** [2], **Ouyang Zhen** [1], **Weilong Yu** [1], **Mohammad Jawad Alami** [3] **and Xuekui Wang** [1,*]

1 College of Plant Science and Technology, Huazhong Agricultural University, Wuhan 430070, China; murtazaalami@webmail.hzau.edu.cn (M.M.A.); pql1226@webmail.hzau.edu.cn (Q.P.); zedangong@webmail.hzau.edu.cn (Z.G.); yangtewu@mail.hzau.edu.cn (T.Y.); oyzzds@webmail.hzau.edu.cn (O.Z.); yuweilong@webmail.hzau.edu.cn (W.Y.)
2 Bureau of Agriculture and Rural Affairs of Lichuan City, Lichuan 445400, China; Tudaiqun@163.com
3 Institute of Urban Environment, Chinese Academy of Sciences, Xiamen 361021, China; jawadalami459@gmail.com
* Correspondence: wang-xuekui@mail.hzau.edu.cn

**Abstract:** Chinese goldthread (*Coptis chinensis* Franch.) represents one of the most important medicinal plants with diverse medicinal applications, but it easily suffers from continuous cropping obstacles in the plantation. In this study, we have selected eight different continuously cropped fields with *C. chinensis* and fallow field, providing detailed information regarding the diversity and composition of the rhizospheric bacterial communities. We have found a significant difference between fallow field (LH) and other continuously cropped fields in soil pH; the total content of nitrogen, phosphorus, and potassium; and soil enzyme activities. The results indicate that continuous cropping had a significant effect on soil physicochemical properties and enzyme activities under different plant cultivations. The relative abundance of bacterial phyla was significantly altered among the fields; for example, *proteobacteria* and *Actinobacteria* were observed to be higher in continuous cropping of maize (HY6) and lower in sweet potato continuous cropping (HH). Alpha diversity analysis showed that different plants with different years of continuous cropping could change the diversity of bacterial communities, among which the effect of maize and Polygonum multiflorum continuous cropping were most significant. Principle coordinate analysis (PCoA) showed that continuously cropped *C. chinensis* (LZ) and cabbage continuously cropped for 2 years (HS) were slightly clustered together and separated from LH and others. The results showed that the similarity of the bacterial community in the same crop rotation was higher, which further indicated that the bacterial community structure was significantly altered by the continuous cropping system and plant species. Our study provides a foundation for future agricultural research to improve microbial activity and increase crops/cash-crops productivity under a continuous cropping system and mitigate continuous cropping obstacles.

**Keywords:** next-generation sequencing; soil bacterial community; continuous cropping; composition; diversity

## 1. Introduction

*Coptis chinensis* Franch belongs to the *Ranunculaceae* family, and it is one of the most important medicinal plants in China [1,2]. It mainly has the function of clearing heat, dampness, purging fire, and detoxification [3–5]. Berberine, epiberberine, palmatine, coptisine, and jatrorrhizine, which are predominantly active isoquinoline alkaloids, have been confirmed as significant constituents in *C. chinensis* [6]. Because of the increase in demand in the market for *C. chinensis* and some other crops and cash crops, these plants have been continuously cultivated in the same field for a long time.

Continuous cropping is a method of replanting plants on soils for a long term. Continuous cropping methods are frequently used in the cultivation of grain crops, cash crops, and medicinal plants due to limited arable areas. Long-term, continuous cropping of plants in the fields increases the continuous cropping obstacles such as decreasing plant growth and yield, the incidence of some serious root rot diseases, decreasing the soil health and quality, and depleting some nutrients from the field [1]. Not only do the plants suffer from continuous cropping, but also the composition, diversity, and structure of microbes are affected by continuous cropping of plants in the field for a long time. Continuous cropping obstacles have been attributed to a variety of reasons, including deterioration of soil physicochemical characteristics, decreases in soil enzymatic activities, accumulation of autotoxic chemicals, and the buildup of soil-borne pathogens. A growing number of studies have hypothesized that disturbance of the soil microbial population leads to continuous cropping obstacles after extended periods of continuous cropping [7].

Rhizosphere microbial communities are closely related to plant growth, nutrient uptake, and the occurrence of soil-borne diseases. Micro-ecology research studies have shown that long-term, continuous cropping can cause significant changes in crop rhizospheric microbial communities [8–10]. The microbial community structure, diversity, and composition are significantly different from fallow or rotation soils [1,2]. Recently, the main methods to repair continuous cropping obstacles are fallow or crop rotation, which can reduce the abundance of harmful microorganisms in the soil. Researchers have improved the technologies to alleviate continuous cropping obstacles from the perspectives of improving the soil microenvironment, changing the composition of microbial communities in the rhizosphere, and increasing the diversity of the microbial community, including the application of organic fertilizers, rotation, or over cropping [11–16]. However, no study has been conducted to compare the effects of continuous cropping on soil physicochemical properties, enzyme activities, and bacterial community among different plants, which is important for the selection of rotational crops for *C. chinensis*.

Thus, this study aims to find the effects that different plant species and continuous cropping have on the soil bacterial community. We hypothesize that different plant/crop species secrete different types of root exudates, which can alter the composition, diversity, and structure of the soil bacterial community. We explored the soil fertility by measuring the soil physicochemical properties, soil enzyme activities, as well as the correlation between soil bacterial community and soil environmental factors under different plant successive cultivation, and we discovered how continuous cropping and plant species impact the rhizosphere soil bacterial community structure and composition. We also give insight on the application of biological control to *C. chinensis* continuous cropping obstacles. This study was conducted in eight different continuous cropping fields with different plants and one fallow field as control, and it was based on a large amount of data from next-generation sequencing technology, such as Illumina Miseq.

## 2. Materials and Methods

### 2.1. Study Site and Soil Sampling

In this experiment, the main production crops in the core production areas of *C. chinensis* were selected and rotated in different periods to explore the effects of different crops and different rotation periods on the soil rhizosphere microorganisms. The soil samples were collected from 8 different continuous cropping fields and 1 fallow field under the same conditions (chemical fertilizer applications, temperature, and light intensity) in Jiannan county, Lichuan City, Hubei province, China (Coordinates: 108°23′–108°35′ E, 30°18′–30°35′ N), altitude 1500 m, soil types mostly sandy and clay, average annual rainfall between 1198 and 1650 mm, and the average yearly temperature is 12.7 °C). In this study, continuously cropped *C. chinensis* (LZ), maize continuously cropped for 2 and 6 years (HY2, HY6), *Polygonum multiflorum* continuously cropped for 2 and 6 years (HW2, HW6), sweet potato continuously cropped for 2 years (HH), *Fritillaria thunbergii* continuously cropped for 2 years (HB), cabbage continuously cropped for 2 years (HS) were used, and fallow field

without cultivating (LH) was the control. Four composite soil samples were taken per field from the roots of 20 randomly selected *C. chinensis*, maize, *P. multiflorum*, sweet potato, *F. thunbergii*, and cabbage plants. The plant roots were shaken vigorously to separate the soil not tightly adhering to the roots. As there were no plants in the fallow field, four composite soil samples consisted of five cores taken about 15 cm in the topsoil. The soil was then transported to the laboratory in iceboxes. In the laboratory, each soil sample was properly sieved to remove debris and other stony materials using a 2 mm sieve. Each sample was appropriately homogenized, and 10 g of soil was put in the sterilized tubes and stored at $-80$ °C for DNA extraction. Other soils were saved for the analysis of soil physicochemical properties and enzyme activities.

### 2.2. Analysis of Soil Physicochemical Properties

The soil pH was measured with a Mettler-Toledo TE 20 (Mettler-Toledo, Columbus, OH, USA) using soil suspension with deionized distilled water (1:20 $w/v$). The soil organic matter was tested using the potassium dichromate internal heating method. The total contents of N and P were measured using a Smartchem 200 Discrete analyzer (Unity Scientific, Milford, MA, USA), and the total content of K was measured using an FP series multielement flame photometer (Xiang Yi, Hunan, China). Soil available nitrogen content was determined by the alkali hydrolyzed diffusion method. Soil available phosphorus content was determined by the sodium bicarbonate extraction molybdenum antimony anti-colorimetry method. Soil available potassium was determined by an ammonium acetate extraction flame photometer. Soil available boron was determined by boiling water extraction curcumin colorimetry [2,17,18].

### 2.3. Analysis of Soil Enzyme Activities

Sucrase activity was measured by 3,5-dinitrosalicylic acid colorimetry, and the results were expressed in mg of glucose produced in 1 g soil after 24 h. Urease activity was measured by indigo blue colorimetry, and the results were expressed in mg of $NH_3$-N in 1 g soil after 24 h. Phosphatase activity was measured using disodium phenyl phosphate [17,18].

### 2.4. DNA Extraction and MiSeq Sequencing

Genomic DNA was extracted from 0.5 g of soil samples (dry weight) using the PowerSoil kit (MO BIO Laboratories, Carlsbad, CA, USA). The concentration of the extracted DNA was determined using a Nanodrop 2000C Spectrophotometer (Thermo Scientific, Wilmington, DE, USA). The DNA extracted from the soil samples was kept at $-80$ °C until it was used.

The V4–V5 regions of the bacterial 16S rRNA gene were amplified with primers 338F (5′-ACTCCTACGGGAGGCAGCAG-3′) and 806R (5′-GGACTACHVG GTWTCTAAT-3′). The following program was used to conduct the PCR reaction: 3 min of denaturation at 95 °C, 27 cycles of 30 s at 95 °C, 30 s for annealing at 55 °C, 45 s for elongation at 72 °C, and a final extension at 72 °C for 10 min.

The PCR reactions were performed in triplicate using a 20 μL mixture containing 4 μL of 5× FastPfu Buffer, 2 μL of 2.5 mM dNTPs, 0.8 μL of each primer (5 μM), 0.4 μL of FastPfu Polymerase, 0.2 μL BSA, and 10 ng of template DNA.

The result of PCR products was extracted from 2% agarose gel, further purified using the AxyPrep DNA Gel Extraction Kit (Axygen Biosciences, Union City, CA, USA), and quantified using QuantiFluor-ST (Promega, Madison, WI, USA) according to the manufacturer's protocol.

### 2.5. Bioinformatic Analyses

We pooled the purified amplicons in equimolar and paired-end sequences (2 × 300) on an Illumina MiSeq platform (Illumina, San Diego, CA, USA) according to standard protocols by the company. Illumina sequencing was used to generate the raw gene sequence data, and then Trimmonmatic and FLASH were used to modify the data. Usearch 7.1 (http:

//www.drive5.com/usearch/, (accessed on 30 October 2021)) [19] was used to cluster the sequences into operational taxonomic units (OTUs) after a 97% pairwise identity by QIIME. Ribosomal Database Project classifier (Release 11.1, http://rdp.cme.msu.edu/, (accessed on 30 October 2021)) [20] was used for the taxonomic classification of the representative sequence for bacteria against the Greengenes 16S rRNA database [21] and Silva [22].

### 2.6. Statistical Analyses

Pearson correlation coefficients between the soil properties and abundances of bacterial phyla were all calculated using SPSS v20.0 (SPSS Inc., Chicago, IL, USA). For alpha diversity, all analyses were based on the OTU clusters with a cutoff of 3% dissimilarity. Chao, Ace, Shannon, and Phylogenetic diversity (Pd) were calculated to estimate the richness and diversity of the bacterial community of each sample. Rarefaction curves with the average number of observed OTUs were generated using Mothur to compare relative levels of bacterial OTU diversity across the 9 different field soils. The weighted and unweighted UniFrac distance metrics (based on the phylogenetic structure) were used to generate PCoA plots to further assess the similarities between the samples' community memberships. Redundancy analysis (RDA) of multiple correlation variations among environmental variables (soil physicochemical properties, soil enzyme activities, and bacterial community composition at the phylum level) was carried out by using Canoco5. In the end, Venn diagrams were constructed to visualize shared and unique species between samples.

## 3. Results

### 3.1. Soil Physicochemical Properties and Enzyme Activities

The changes in soil pH in different fields are shown in Figure 1. Both maize continuous cropping and *F. thunbergii* continuous cropping could increase soil pH, and it was significantly higher than that of LH ($p < 0.05$). The other fields were significantly lower than the LH. In the same plant species, there was no significant difference in soil pH between maize continuous cropping and *P. multiflorum* continuous cropping fields with the increase in cultivation years. The results indicate that the soil pH was different in different plant cultivated fields and was directly affected by the plant species.

Soil organic matter content was different in different continuous cropping fields, which is shown in Figure 2. The soil organic matter content of LH was significantly (51.46%) higher compared to the others. The lowest organic matter content was observed in HW2 followed by HS. In the same plant species, the soil organic matter content of maize fields decreased with the increase in cultivation year, while the rotation of *P. multiflorum* was significantly increased.

The changes of soil available nitrogen, phosphorus, potassium, and boron are shown in Table 1. The soil available nitrogen content of LH was higher compared with the other continuous cropping fields. In the same plant species, the soil available nitrogen content in maize continuous cropping fields significantly decreased with the increase in the cultivation years, but in the *P. multiflorum* continuous cropping fields it increased. The fallow field (LH) had the lowest available phosphorus among the fields, and the differences were significant. The soil available potassium content was different in different continuous cropping fields. The maize continuous cropping fields showed an increasing trend with the increase in cropping years, and the other fields showed a decreasing trend. Fallow field (LH) had lower available potassium after HW2, HW6, and HH. The changes of soil available boron content in different continuous cropping fields were different, and the lowest content was observed in LH. In the same plant species, the soil available boron content in maize rotation significantly decreased with the increase in the years, while for *P. multiflorum* there was no significant difference in the soil available boron content with the increase in the years.

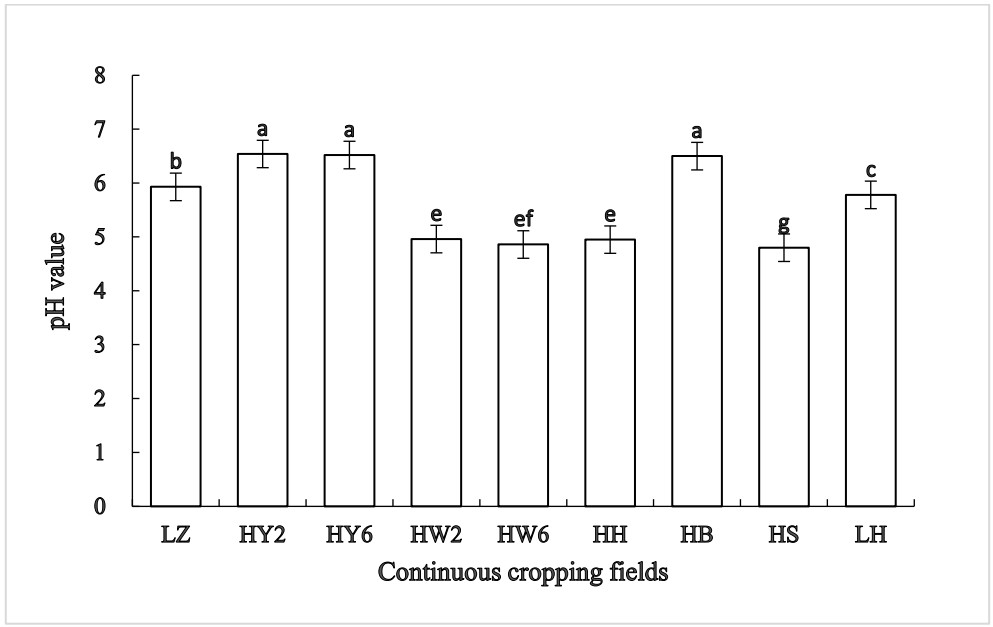

**Figure 1.** pH value in different continuous cropping fields. LZ, HY2, HY6, HW2, HW6, HH, HB, HS, and LH; continuous cropping of *C. chinensis* for 5 years, maize continuous cropping for 2 years, maize continuous cropping for 6 years, *P. multiflorum* continuous cropping for 2 years, *P. multiflorum* continuous cropping for 6 years, sweet potato continuous cropping for 2 years, *F. thunbergia* continuous cropping for 2 years, cabbage continuous cropping for 2 years, and fallow field, respectively. Different letters indicate significant differences between the fields (*p*-Value < 0.05).

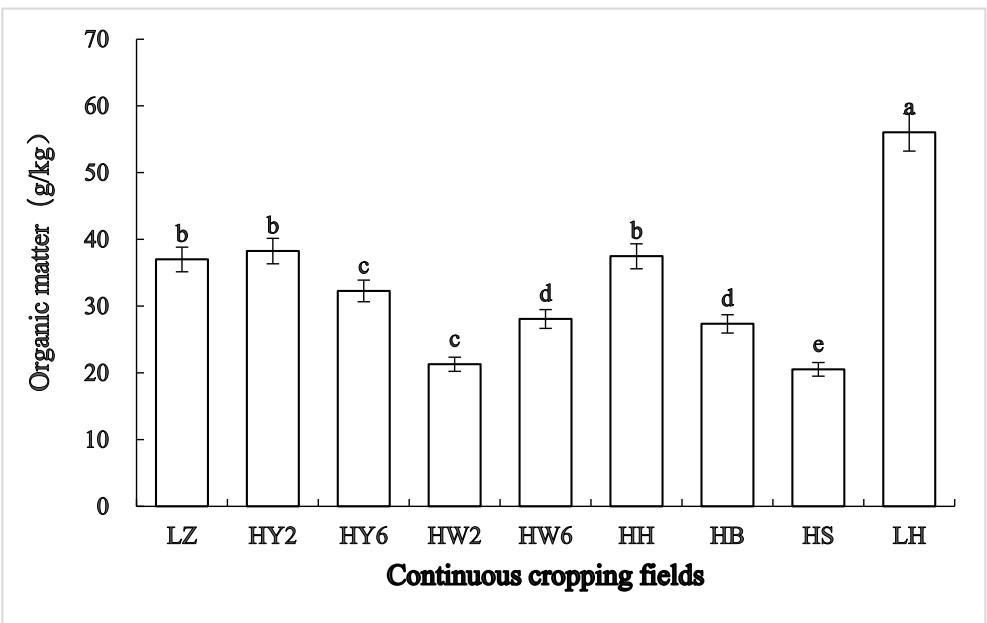

**Figure 2.** Content of organic matter under different continuous cropping and fallow fields. LZ, HY2, HY6, HW2, HW6, HH, HB, HS, and LH; continuous cropping of C. chinensis for 5 years, maize continuous cropping for 2 years, maize continuous cropping for 6 years, *P. multiflorum* continuous cropping for 2 years, *P. multiflorum* continuous cropping for 6 years, sweet potato continuous cropping for 2 years, F. thunbergia continuous cropping for 2 years, cabbage continuous cropping for 2 years, and fallow field, respectively. Different letters indicate significant differences between the fields (*p*-Value < 0.05).

**Table 1.** Soil available N, P, K, and boron in different continuous cropping fields.

| Fields | Available Nitrogen (mg/kg) | Available Phosphorus (mg/kg) | Available Potassium (mg/kg) | Available Boron (mg/kg) |
|---|---|---|---|---|
| LZ | 118.98 ± 0.01 d | 179.84 ± 0.39 b | 204.31 ± 1.20 c | 0.20 ± 0.03 cd |
| HY2 | 132.97 ± 0.03 b | 127.56 ± 0.58 e | 583.45 ± 0.50 a | 0.39 ± 0.02 a |
| HY6 | 118.95 ± 0.01 d | 72.92 ± 6.84 f | 514.95 ± 2.03 b | 0.24 ± 0.00 bc |
| HW2 | 69.97 ± 0.00 i | 36.44 ± 0.75 g | 53.33 ± 0.88 h | 0.17 ± 0.00 ef |
| HW6 | 90.98 ± 0.01 g | 22.33 ± 0.33 h | 73.66 ± 1.20 g | 0.19 ± 0.01 de |
| HH | 125.97 ± 0.01 c | 297.05 ± 1.12 a | 54.99 ± 0.58 h | 0.27 ± 0.02 b |
| HB | 97.99 ± 0.01 f | 165.48 ± 0.58 c | 129.65 ± 0.66 e | 0.23 ± 0.01 bcd |
| HS | 104.97 ± 0.02 e | 154.16 ± 2.83 d | 157.65 ± 0.34 d | 0.24 ± 0.01 bc |
| LH | 139.98 ± 0.00 a | 3.76 ± 0.06 i | 110.65 ± 1.76 f | 0.13 ± 0.01 f |

Continuous cropping of *C. chinensis* for 5 years, maize continuous cropping for 2 years, maize continuous cropping for 6 years, *P. multiflorum* continuous cropping for 2 years, *P. multiflorum* continuous cropping for 6 years, sweet potato continuous cropping for 2 years, *F. thunbergia* continuous cropping for 2 years, cabbage continuous cropping for 2 years, and fallow field; LZ, HY2, HY6, HW2, HW6, HH, HB, HS, and LH, respectively. Different letters indicate significant differences between the fields (*p*-Value < 0.05).

As shown in Table 2, HY6, HH, and LH fields had a significantly higher total content of nitrogen, while HW2 had a lower total content of nitrogen. In the same plants, the total content of nitrogen increased with the increase in the cultivation years of maize and *P. multiflorum*, but the differences were not significant. The total content of phosphorus was higher in HY2, HH, and HB and lowest in LH. The total content of phosphorus showed a significant decreasing trend with the increase in the cultivation years in maize, while *P. multiflorum* showed an increasing trend. The total content of potassium was increased with the increase in the cultivation years of maize fields, while in *P. multiflorum* it significantly decreased. In LH the total content of potassium was lower after HW2, HW6, and HH fields.

**Table 2.** Soil total content of nitrogen, phosphorus, and potassium in different continuous cropping fields.

| Fields | Total Nitrogen (g/kg) | Total Phosphorus (g/kg) | Total Potassium (g/kg) |
|---|---|---|---|
| LZ | 0.92 ± 0.01 abc | 12.87 ± 0.08 d | 7.98 ± 0.07 c |
| HY2 | 0.90 ± 0.05 abc | 17.19 ± 0.14 b | 7.28 ± 0.02 d |
| HY6 | 0.95 ± 0.04 ab | 12.53 ± 0.10 de | 12.75 ± 0.09 a |
| HW2 | 0.46 ± 0.02 d | 4.09 ± 0.08 h | 5.64 ± 0.03 f |
| HW6 | 0.76 ± 0.05 c | 4.82 ± 0.10 g | 3.67 ± 0.07 i |
| HH | 1.06 ± 0.07 a | 20.84 ± 0.20 a | 5.06 ± 0.11 g |
| HB | 0.92 ± 0.04 abc | 15.95 ± 0.15 c | 12.21 ± 0.05 b |
| HS | 0.77 ± 0.01 bc | 12.45 ± 0.06 e | 12.43 ± 0.30 ab |
| LH | 1.04 ± 0.10 a | 3.85 ± 0.09 h | 6.34 ± 0.10 e |

Continuous cropping of *C. chinensis* for 5 years, maize continuous cropping for 2 years, maize continuous cropping for 6 years, *P. multiflorum* continuous cropping for 2 years, *P. multiflorum* continuous cropping for 6 years, sweet potato continuous cropping for 2 years, *F. thunbergia* continuous cropping for 2 years, cabbage continuous cropping for 2 years, and fallow field; LZ, HY2, HY6, HW2, HW6, HH, HB, HS, and LH, respectively. Different letters indicate significant differences between the fields (*p*-Value < 0.05).

As shown in Table 3, the soil urease activity compared with LH was significantly higher in maize continuous cropping fields (HY2 and HY6). In the same plant species, the soil urease activity significantly increased with the increase in the years of maize and *P. multiflorum* fields. Soil sucrase activity was different in different fields. The lower content was in LH after HH, HB, and HS. Among the various continuous cropping and fallow fields, the HY6 field was more conducive to the increase in sucrase activity. The soil sucrase activity of maize continuous cropping fields significantly increased with the increase in the cultivation years, while the *P. multiflorum* showed a decreasing trend. Soil phosphatase activity was higher in HS followed by LH and lowest in HH. In maize fields, the phosphatase activity increased with the increase in the cropping years, while in the *P. multiflorum* fields, phosphatase activity significantly decreased.

**Table 3.** Soil enzyme activities in different continuous cropping fields.

| Fields | Urease Activities (mg/(g·d)) | Sucrase Activities (mg/(g·d)) | Phosphatase Activities (mg/(g·d)) |
|---|---|---|---|
| LZ | 0.65 ± 0.00 f | 20.82 ± 0.34 b | 3.09 ± 0.02 def |
| HY2 | 1.38 ± 0.01 b | 19.43 ± 0.48 c | 3.53 ± 0.06 bc |
| HY6 | 1.85 ± 0.01 a | 32.76 ± 0.04 a | 4.03 ± 0.11 a |
| HW2 | 0.75 ± 0.02 e | 15.52 ± 0.21 d | 3.12 ± 0.04 de |
| HW6 | 0.85 ± 0.01 d | 13.39 ± 0.16 e | 2.85 ± 0.03 f |
| HH | 0.79 ± 0.04 e | 8.39 ± 0.10 g | 2.46 ± 0.18 g |
| HB | 0.88 ± 0.01 d | 8.69 ± 1.12 g | 3.72 ± 0.02 b |
| HS | 0.86 ± 0.01 d | 7.29 ± 0.16 h | 2.96 ± 0.03 ef |
| LH | 0.99 ± 0.02 c | 10.92 ± 0.21 f | 3.65 ± 0.06 ab |

Continuous cropping of *C. chinensis* for 5 years, maize continuous cropping for 2 years, maize continuous cropping for 6 years, *P. multiflorum* continuous cropping for 2 years, *P. multiflorum* continuous cropping for 6 years, sweet potato continuous cropping for 2 years, *F. thunbergia* continuous cropping for 2 years, cabbage continuous cropping for 2 years, and fallow field; LZ, HY2, HY6, HW2, HW6, HH, HB, HS, and LH, respectively. Different letters indicate significant differences between the fields (*p*-Value < 0.05).

*3.2. Bacterial Community Composition and Structure Variations*

The bacterial community V3-V4 region of 16S RNA in different soil samples was sequenced and classified. A total of 2 kingdoms, 57 phyla, 164 classes, 403 orders, 640 families, 1242 genera, and 2900 species were detected in 36 groups of test soils. Among the two kingdoms in the soil, the bacterial kingdom was 97.24% and the archaea kingdom 2.76%. The soil bacterial kingdom mainly included 16 bacterial phyla (as shown in Figures 3A and S1B), which were *Proteobacteria* (33.1%), *Actinobacteria* (17.61%), *Acidobacteriota* (14.02%), *Chloroflexi* (12.97%), *Firmicutes* (3.02%)), *Gemmatimonadota* (2.9%), *Bacteroidota* (2.59%), and *Crenarchaeota* (2.52%), accounting for 88.78% of the total number of bacterial phyla. *Proteobacteria* and *Actinobacteria* were the two most dominant phyla in the bacterial community. At the genus level, no-rank AD3 class and *bradyrhizobium* were the most dominant genera in the bacterial community (Figure S1A).

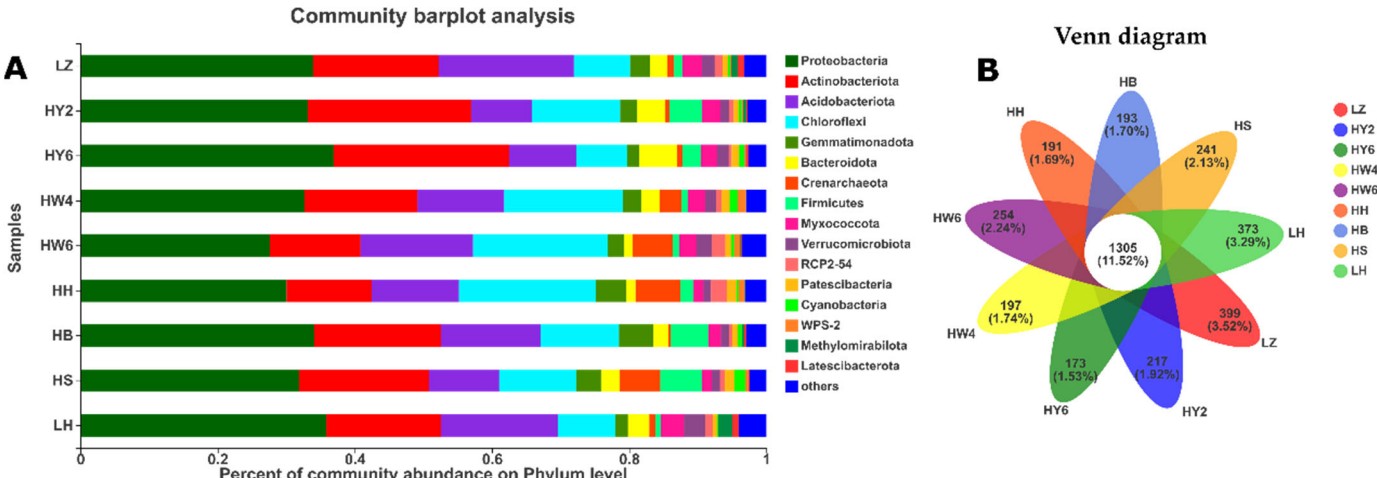

**Figure 3.** Bacterial composition and structure under different rotational cropping patterns. (**A**) Bacterial community barplot at the phylum level. (**B**) Venn diagram of unique and shared bacterial species among the fields. LZ, HY2, HY6, HW2, HW6, HH, HB, HS, and LH; continuous cropping of *C. chinensis* for 5 years, maize continuous cropping for 2 years, maize continuous cropping for 6 years, *P. multiflorum* continuous cropping for 2 years, *P. multiflorum* continuous cropping for 6 years, sweet potato continuous cropping for 2 years, *F. thunbergia* continuous cropping for 2 years, cabbage continuous cropping for 2 years, and fallow field, respectively.

As shown in Figure 3B, the number of species shared by the soil samples between different continuous cropping fields was 1305, accounting for 11.52%. The number of

unique species in the LZ field was the largest among the fields (399), accounting for 3.52%, and was followed by the LH (373), accounting for 3.29%. In the same plant species, the number of unique species in maize decreased with the increase in the years, while in *P. multiflorum* it increased. The results show that the order of the number of unique species in each group was LZ > LH > HW6 > HS > HY2 > HW4 > HB > HH > HY6, with 399, 373, 254, 241, 217, 197, 193, 191, and 173, respectively, indicating that continuous cropping of different plant species had a significant influence on unique soil species of the bacterial community.

One-way ANOVA results showed that the relative abundance of *Proteobacteria* had no significant difference among the continuous cropping fields, but a higher abundance was observed in HY6 (36.89%), *Actinobacteria* was also higher in HY6 (25.62%) and had the lowest abundance in HH (12.37%). *Acidobacteriota* had the highest abundance in LZ (19.75%) and the lowest abundance in HY4 (8.79%). Other bacterial phyla were significantly changed by selecting different plant species. In the same species, the bacterial phyla were significantly different and showed an increase in *Proteobacteria*, *Actinobacteria*, and *Acidobacteriota* relative abundance by the increasing in cropping years in maize continuous cropping fields, while in *P. multiflorum* there was a significant decreasing trend (Figure 4).

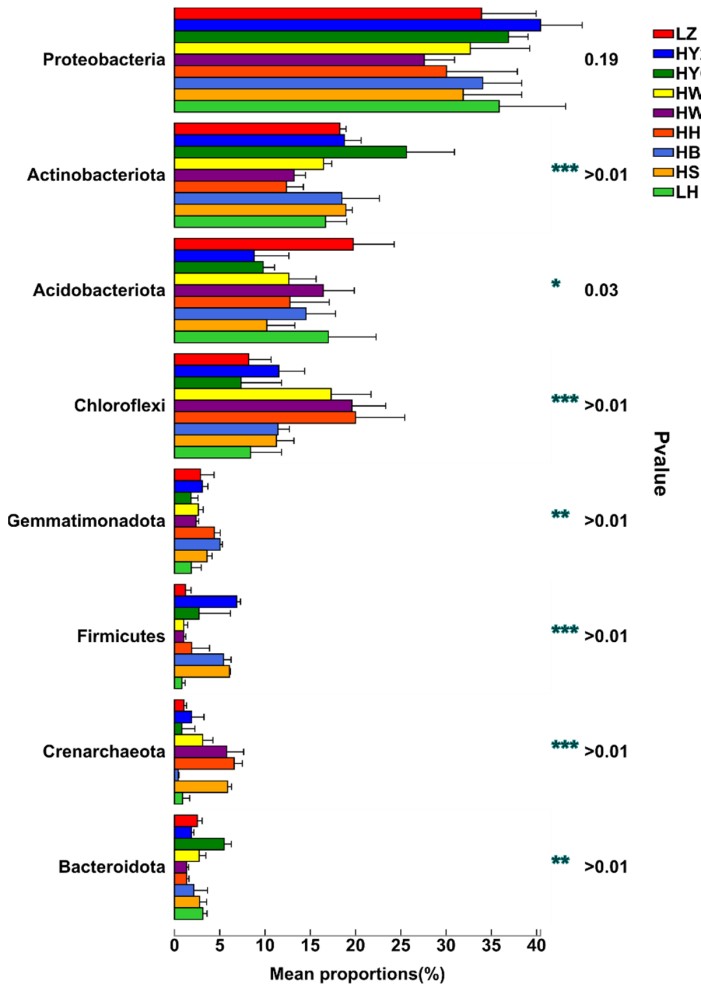

**Figure 4.** Multiple comparisons of soil bacterial community at the phylum level. LZ, HY2, HY6, HW2, HW6, HH, HB, HS, and LH; continuous cropping of *C. chinensis* for 5 years, maize continuous cropping for 2 years, maize continuous cropping for 6 years, *P. multiflorum* continuous cropping for 2 years, *P. multiflorum* continuous cropping for 6 years, sweet potato continuous cropping for 2 years, *F. thunbergia* continuous cropping for 2 years, cabbage continuous cropping for 2 years, and fallow field, respectively.

### 3.3. Microbial Diversity

3.3.1. Alpha Diversity

The 16S rRNA Illumina Miseq sequencing technology was used to sequence the rhizosphere soil samples of *C. chinensis* in different rotation patterns. Four replicate samples were set for each field, for a total of 36 samples. A total of 1,508,600 valid sequences were obtained from the samples, with an average length of 413 bp. These sequences were clustered into 11,609 OTUs with a similarity of 97%. As shown in Figure S2, the rarefaction curve of each sample tended to be flat as the amount of sequencing data increased, indicating that the amount of sequencing data was large enough to reflect most of the bacterial species' information in the sample. The sequencing coverage of each sample was greater than 95%, indicating that the sample sequencing result already contained most of the bacterial species in the sample.

As shown in Figure 5, a lower Shannon index value was observed in HH and HS fields, and a higher index value was in HY2, HY6, and HW6. The difference between LH and other fields was not significant (*p*-Value > 0.05). In the same species, the Shannon value of maize continuous cropping fields decreased with the increase in the years, while it increased in *P. multiflorum* continuous cropping fields. Chao index showed a lower value in HH and a higher value in other fields, and the highest value of the Sobs index was observed in the HY2. In the maize and *P. multiflorum* continuous cropping fields, Sobs and Chao1 indices showed a significant decreasing trend with the increase in cropping years. The Pd index value was lower in the HH and higher in other cropping fields. Among them, HY2, HY6, and HB increased greatly, but only the difference in the HY2 field was significant (*p* < 0.05), indicating that two-year continuous cropping of maize can increase the diversity of the bacterial community. Alpha diversity analysis showed that different plant species could change the diversity of bacterial communities; among them the effect of maize and *P. multiflorum* were the most prominent.

3.3.2. Beta Diversity

The results of principal coordinate analysis (PCoA) of soil samples from different continuous cropping fields are shown in Figure 6. The cumulative contribution rates of the two maximally reflected differences among fields were 26.49% and 18.08%, respectively. The differences caused by different plant species are along the axis of PC1 and PC2. From the perspective of the PC1 axis, LZ was distributed on both positive and negative axes, while HY2, HY6, HB, and HS were on the positive axis, and HW2, HW6, HH, and LH were on the negative axis. From the perspective of the PC2 axis, LZ and LH were on the positive axis, and the rest were on the negative axis. The analysis showed the fallow fields (LH) separated from the other cultivated fields. This result showed that the similarity of the bacterial community in the same plant species was higher, which further indicated that the bacterial community structure was significantly altered by the continuous cropping and plant species.

Hierarchical clustering analysis showed the same result as the principal coordinate analysis (PCoA). The bacterial community in the same plants was more similar than in different plants' continuous cropping fields. The soil samples from the LH fields clustered together and were located in the same clade with LZ fields (Figure S3).

### 3.4. Effect of Environmental Factors on Bacterial Communities

Based on the RDA analysis of dominant bacteria and soil physicochemical properties under different rotational cropping patterns, the two-dimensional sequence diagram of RDA (Figure 7B) was obtained. It can be seen from the diagram that the interpretations of the first axis and the second axis were 32.37% and 15.22%, respectively. As shown in Figure 7B, the arrow lines of available potassium, total potassium, and pH value were longer, which indicates that available potassium, total potassium, and pH value had strong explanatory power for the variation of dominant bacterial phyla. Three of them had the same direction and small angle with *Actinobacteria*, *Proteobacteria*, and *bacteroidota*,

showing a significantly positive correlation, and *Chloroflex*, *acidobacteria*, *verrucomicrobiota*, and *gemmatimonadota* had a positive correlation but were not significant. It can be seen from Table S1 that the effects of TP, AN, pH, AK, TK, and AB on the dominant species reached a significant level ($p < 0.01$), indicating that soil nitrogen, phosphorus, potassium, and soil pH affected the composition of the soil bacterial community.

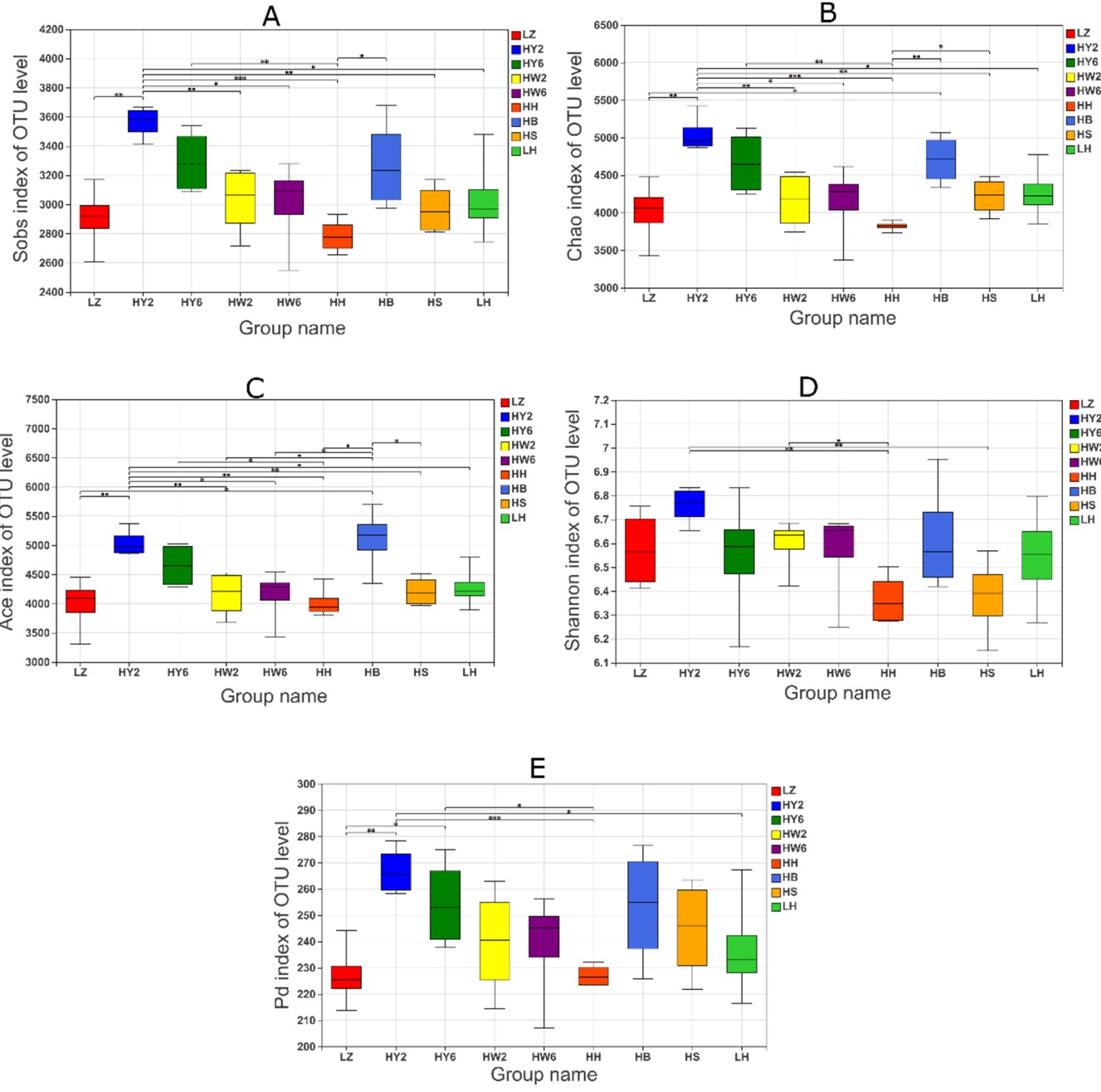

**Figure 5.** Alpha diversity of the bacterial community. (**A**) The number of observed species (Sobs), (**B**) Chao indices, (**C**) Ace indices, (**D**) Shannon indices, and (**E**) phylogenetic diversity indices (Pd). LZ, HY2, HY6, HW2, HW6, HH, HB, HS, and LH; continuous cropping of *C. chinensis* for 5 years, maize continuous cropping for 2 years, maize continuous cropping for 6 years, *P. multiflorum* continuous cropping for 2 years, *P. multiflorum* continuous cropping for 6 years, sweet potato continuous cropping for 2 years, *F. thunbergia* continuous cropping for 2 years, cabbage continuous cropping for 2 years, and fallow field, respectively. Asterisks (*, **, and ***) show significant differences at *p*-value < 0.05, 0.01, and 0.001, respectively.

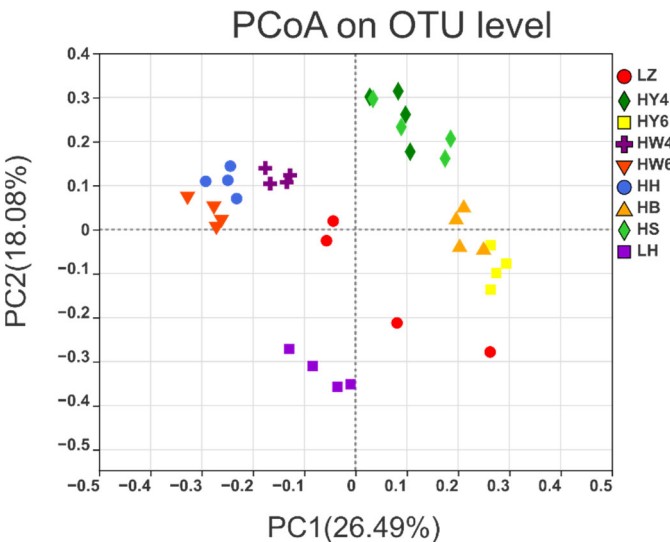

**Figure 6.** Principle coordinate analysis (PCoA) of the bacterial community. LZ, HY2, HY6, HW2, HW6, HH, HB, HS, and LH; continuous cropping of *C. chinensis* for 5 years, maize continuous cropping for 2 years, maize continuous cropping for 6 years, *P. multiflorum* continuous cropping for 2 years, *P. multiflorum* continuous cropping for 6 years, sweet potato continuous cropping for 2 years, *F. thunbergia* continuous cropping for 2 years, cabbage continuous cropping for 2 years, and fallow field, respectively.

The Spearman's correlation coefficient between bacterial phyla and soil physicochemical properties under different continuous cropping fields is shown in Figure 7D. The results showed that some of the dominant bacterial phyla were strongly correlated with soil physicochemical properties, and all the physicochemical properties were positively correlated with *Actinobacteriota*, *Proteobacteria*, *Bacteroidota*, *Firmicutes*, and *Nitrospirota*. pH, AK, and TK were negatively correlated with RCP2-54, *Chloroflex*, WPS-2, *Acidobaracterita*, *Verrucomicrobiota*, and *Cyanobacteria*. AB, AP, and TP were positively correlated with *Firmicutes* and *Nitrospirota*, while they were negatively correlated with *Myxococcota* and *Verrucomicrobiota*. OM, AN, and TN were negatively correlated with WPS-2 and *Cyanobacteria*.

As shown in Figure 7A, the arrow lines of urease and phosphatase were long, which indicates that urease and phosphatase had a strong influence on the variation of dominant bacterial phyla, and they were in the same direction as *Actinobacteria*, *bacteroidota*, *Proteobacteria*, *Firmicutes*, and *Myxococcota*, with a small angle showing a significantly positive correlation. According to Table S2, the effects of urease, phosphatase, and sucrase on the dominant bacterial phyla were significant (*p*-Value < 0.01), indicating that the soil enzyme activities affected the composition and structure of the soil bacterial community.

The correlation analysis of bacterial phyla with soil enzyme activity under different continuous cropping fields is shown in Figure 7C. The results showed that soil urease, sucrase, and phosphatase had a significant and positive correlation with *Actinobacteria*, *bacteroidota*, *Proteobacteria*, *Myxococcota*, *Chloroflex*, *Gemmatimonadota*, RCP2-54, and WPS-2. Soil enzyme activities had different degrees of correlation with dominant bacterial phyla, which indicated that enzyme activities are one of the significant factors for the enrichment of dominant bacteria.

### 3.5. Analysis of Functional Bacteria under Different Continuous Cropping Fields

For the analysis of functional bacterial genera, we selected the 23 most dominant bacterial genera among all annotated genera with the function of nitrogen and carbon cycling. According to the functions of all annotated bacterial genera, the relative abundance of most functional bacteria under different continuous cropping fields was significantly different. Sphingomonas is a large group of Gram-negative, aerobic bacteria with diverse functions in the soil such as biocontrol, degradation of aromatic compounds, dissolution of

phosphorus, and resistance to a variety of pathogens. Nitrogen fixation was significantly higher in cultivated fields than in fallow fields. The highest relative abundance was observed in HB, and the lowest was observed in LH. Among the genera involved in N cycling, *Bradyrhizobium* and *Devosia* with symbiotic nitrogen fixation and nitrate denitrification functions were higher in LH (control), and *Nocardioides* with the nitrate reduction function was higher in the maize continuous cropping field for 6 years (HY6). In carbon cycling bacterial genera, the chemotrophic aerobic bacteria Reyranella were significantly higher in the fallow field and had a lower abundance in HH. The lowest relative abundance of *Sphingobium* and Blastococcus, with the functions of degradation of aromatic compounds and lignin and control of soil-borne diseases, respectively, were observed in LH (control). *Arenimonas*, *Gaiella*, and *Rhodanobacter* with the function of nitrogen reduction were lower in the fallow field. The relative abundance of bacterial genera involved in soil C and N cycling, thereby beneficial to soil nutrient transformation, was greatly improved by maize continuous cropping for 2 and 6 years (Table 4).

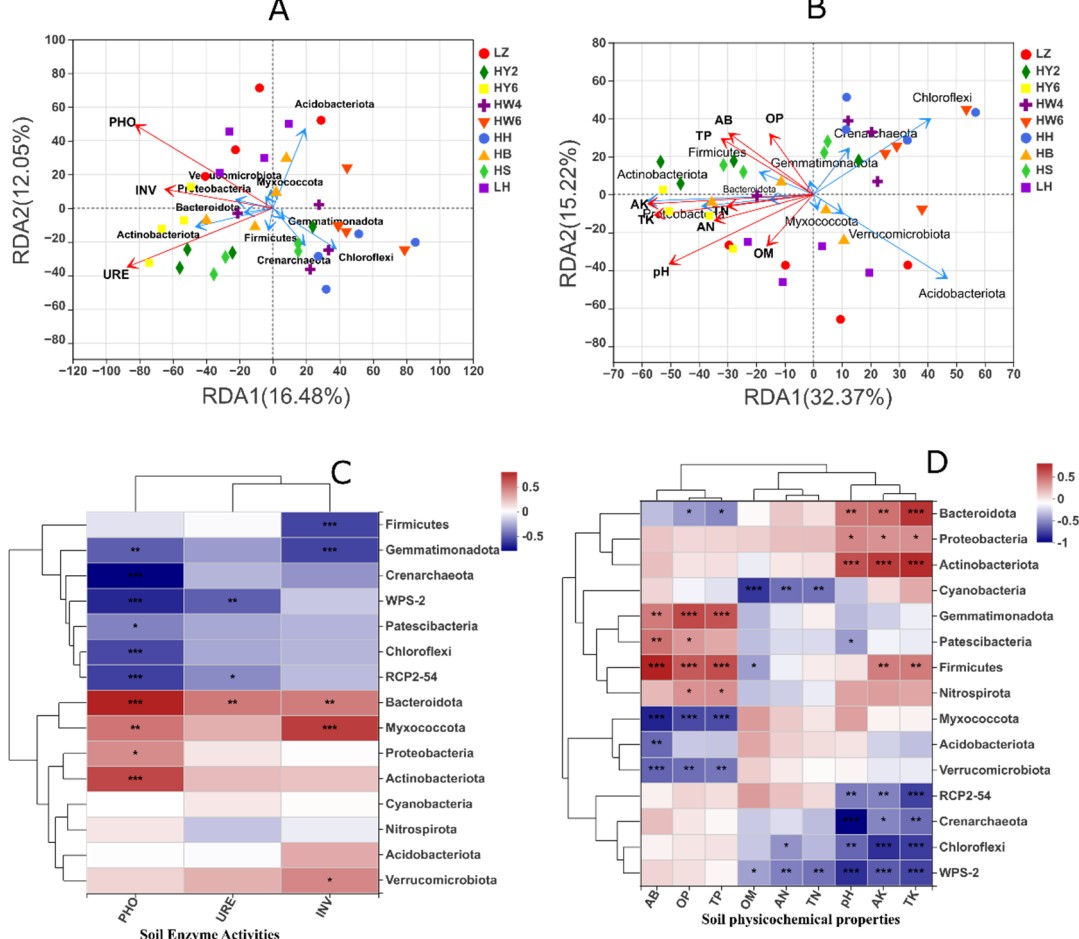

**Figure 7.** The correlation between bacterial phyla, soil physicochemical properties, and soil enzyme activities. (**A**) Redundancy analysis (RDA) of the dominant phyla and soil enzyme activities. (**B**) Redundancy analysis (RDA) of the dominant phyla and soil physiochemical properties. (**C**) Spearman's correlation coefficient of soil enzyme activities and bacterial phyla. (**D**) Spearman's correlation coefficient of soil physicochemical properties and bacterial phyla. LZ, HY2, HY6, HW2, HW6, HH, HB, HS, and LH; continuous cropping of *C. chinensis* for 5 years, maize continuous cropping for 2 years, maize continuous cropping for 6 years, *P. multiflorum* continuous cropping for 2 years, *P. multiflorum* continuous cropping for 6 years, sweet potato continuous cropping for 2 years, *F. thunbergia* continuous cropping for 2 years, cabbage continuous cropping for 2 years, and fallow field, respectively. The length of the species arrow indicates the proportion of species change, and the length of the physicochemical properties arrow indicates the impact on the dominant phylum; the longer the arrow, the greater the impact. Asterisks (*, **, and ***) show significant correlation at *p*-value < 0.05, 0.01, and 0.001, respectively.

**Table 4.** Relative abundance of functional bacterial genera in different continuous cropping fields.

| | Bacterial Genera | Functions | Relative Abundance (%) | | | | | | | | |
|---|---|---|---|---|---|---|---|---|---|---|---|
| | | | LZ | HY2 | HY6 | HW2 | HW6 | HH | HB | HS | LH |
| N | *Sphingomonas* | Biocontrol bacteria, degradation of aromatic compounds, dissolution of phosphorus, resistance to a variety of pathogens, nitrogen fixation | 1.07 | 1.98 | 2.22 | 1.84 | 0.79 | 0.55 | 2.33 | 1.87 | 0.38 |
| | *Gemmatimonas* | Nitrogen fixation | 0.82 | 0.86 | 0.9 | 1.45 | 0.63 | 0.64 | 1.49 | 1.2 | 0.49 |
| | *Bradyrhizobium* | Symbiotic nitrogen fixation and nitrate denitrification | 2.73 | 3.07 | 3.33 | 3.47 | 2.64 | 2.29 | 3.08 | 2.13 | 4.21 |
| | *Devosia* | Symbiotic nitrogen fixation | 0.25 | 0.51 | 0.46 | 0.52 | 0.11 | 0.12 | 0.36 | 0.58 | 0.08 |
| | *Nocardioides* | Nitrate reduction | 0.4 | 0.77 | 1.07 | 0.05 | 0.03 | 0.03 | 0.49 | 0.23 | 0.22 |
| | *Pseudonocardia* | Nitrate reduction | 0.13 | 0.18 | 0.13 | 0.12 | 0.09 | 0.1 | 0.15 | 0.06 | 0.13 |
| | *Rhodopseudomonas* | Nitrogen fixation | 0.04 | 0.12 | 0.08 | 0.08 | 0.02 | 0.03 | 0.11 | 0.13 | 0.08 |
| C | *Reyranella* | Chemotrophic aerobic bacteria | 0.47 | 0.56 | 0.46 | 0.5 | 0.38 | 0.3 | 0.63 | 0.39 | 0.65 |
| | *Edaphobacter* | Chemotrophic aerobic bacteria | 0.08 | 0.04 | 0.1 | 0.03 | 0.05 | 0.01 | 0.04 | 0.04 | 0.03 |
| | *Sphingobium* | Degradation of aromatic compounds and lignin | 0.04 | 0.1 | 0.15 | 0.01 | 0 | 0.01 | 0.02 | 0.03 | 0 |
| | *Blastococcus* | Control of soil borne diseases | 0.05 | 0.27 | 0.26 | 0.58 | 0.2 | 0.11 | 0.13 | 0.3 | 0.01 |
| C,N | *Arenimonas* | Nitrate reduction | 0.04 | 0.24 | 0.31 | 0.06 | 0.01 | 0 | 0.1 | 0.1 | 0.04 |
| | *Gaiella* | Nitrate reduction | 0.67 | 1.49 | 0.68 | 0.16 | 0.25 | 0.39 | 0.85 | 0.27 | 0.55 |
| | *Rhodanobacter* | Nitrate reduction | 0.5 | 0.9 | 0.73 | 2.72 | 0.48 | 0.58 | 1.01 | 2.45 | 0.14 |
| | *Mesorhizobium* | Symbiotic nitrogen fixation | 0.22 | 0.57 | 0.51 | 0.61 | 0.18 | 0.13 | 0.37 | 0.47 | 0.04 |
| | *Terrabacter* | Degrading bacteria and nitrate reduction | 0.03 | 0.74 | 1.12 | 0.27 | 0.04 | 0.02 | 0.36 | 0.88 | 0 |
| | *Arthrobacter* | Degrading bacteria, desulfurization, phosphorus dissolution, nitrate reduction, lignin decomposition | 0.61 | 3.54 | 3.82 | 1.41 | 0.62 | 0.2 | 1.71 | 2.98 | 0.07 |
| Others | *Micromonospora* | Degrading bacteria and producing a variety of antibiotics | 0.17 | 0.3 | 0.11 | 0.11 | 0.12 | 0.06 | 0.15 | 0.06 | 0.13 |
| | *Actinoplanes* | Produce a variety of antibiotics | 0.17 | 0.22 | 0.27 | 0.33 | 0.28 | 0.07 | 0.16 | 0.07 | 0.06 |
| | *Mycobacterium* | Degrading bacteria, nitrogen removal and phosphorus dissolution | 0.49 | 0.73 | 0.92 | 0.69 | 0.77 | 0.61 | 0.88 | 0.56 | 0.59 |
| | *Microbacterium* | Desulphurization | 0.15 | 0.45 | 0.32 | 0.06 | 0 | 0.01 | 0.06 | 0.15 | 0.03 |
| | *Lysobacter* | Biocontrol bacteria | 0.1 | 0.3 | 0.55 | 0.02 | 0.01 | 0 | 0.07 | 0.1 | 0.02 |
| | *Streptomyces* | Produce antibiotics | 0.26 | 0.29 | 0.42 | 0.24 | 0.15 | 0.06 | 0.26 | 0.07 | 0.09 |

Continuous cropping of *C. chinensis* for 5 years, maize continuous cropping for 2 years, maize continuous cropping for 6 years, *P. multiflorum* continuous cropping for 2 years, *P. multiflorum* continuous cropping for 6 years, sweet potato continuous cropping for 2 years, *F. thunbergia* continuous cropping for 2 years, cabbage continuous cropping for 2 years, and fallow field; LZ, HY2, HY6, HW2, HW6, HH, HB, HS, and LH, respectively. The numbers show the relative abundance (proportion %) of the bacterial community at the genus taxonomic level.

## 4. Discussion

Understanding the soil physicochemical properties and soil enzyme activities in different plant continuous cropping fields could help us better understand the soil productivity and fertility in different plant and *C. chinensis* continuous cropping fields. In this study, we found that both maize and *F. thunbergii* continuous cropping (HY2, HY6, and HB) increased soil pH, which was significantly higher than that of the *C. chinensis* continuous cropping field (*p*-Value < 0.05); HW2, HW6, HH, HS, and LH were significantly lower than *C. chinensis* continuous cropping soil. Similarly, Xiong et al. (2015) reported that the continuous cropping of black pepper decreased the soil pH [18], and similar results also have been reported in other works [23–25]. We have found a higher content of soil organic matter in the fallow field (LH). This means that different plant species changed the content of organic matter. Grandy et al. (2002) also reported that the soil organic matter was changed by potato continuous cropping [26]. In this study, soil available nitrogen was higher in HY2 and HH, soil available phosphorus was higher in HH, and soil available potassium was higher in both maize continuous cropping fields (HY2 and HY6). It shows that selecting the plant had a significant influence on soil nutrient availabilities. These results were inconsistent with many other findings [27–29]. Maize continuous cropping for 6 years (HY6) and sweet potato continuous cropping for 2 years (HH) showed an increasing trend in the total content of nitrogen, while other fields showed a decreasing trend. The total content of phosphorus significantly increased in HY2, HH, and HB, and total content of potassium increased in HY6, HB, and HS. These results indicate that continuous cropping increased the total content of N, P, and K in soil. Similar results have also been observed in maize [30], potato [31], and wheat [32]. At present, most crops are harmed by continuous cropping, resulting in significant changes in soil enzyme activity. Soil transformation efficiency is inhibited, and accumulation is affected [12,26,31,32]. In this study, we have found that continuous cropping can effectively increase soil enzyme activity in different plant species, which is conducive to the conversion of soil substances and normal crops. Soil urease activity in cultivated fields significantly increased. The HY6 field had the highest soil sucrose activity, and soil phosphatase in HY2, HY6, and HB showed significant increases. A similar result was also reported by Xiao et al. (2015) in *C. morifolium* [33].

Changing the diversity and composition of rhizosphere soil microorganisms has a considerable influence on soil health and fertility. Soil microorganisms play an essential role in maintaining plant health, and the associated microbial community is also introduced as the second genome of the plant [34–38]. However, according to our findings, continuous cropping had a significant influence on the composition, diversity, and structure of the bacterial community in soil [39–41]. In this study, compared with LH (control), Shannon indices were higher in HY2, HY6, and HW6 and lower in HH and HS. Sobs, Chao1, and Pd indices were higher in HH and lower in other fields. A similar work also was reported, in which the soil microbial diversity had been almost completely eradicated after five years of continuous cropping, and a higher value was found in the one-year-old than in the five-year-old *C. chinensis* plant [1]. Similar results have also been reported in cucumber [42] and potato [43], where the cultivation method and practices significantly changed the diversity of the bacterial community in the rhizospheric soil. The difference among the species was significant, which means that the plant species and continuous cropping could change the bacterial community's diversity compared with the fallow field [44,45].

*Proteobacteria*, *Actinobacteria*, *Acidobacteriota*, and *Chloroflexi* were the four most dominant bacterial phyla. *Proteobacteria* plays an important role in maintaining the function of the soil ecosystem, providing help for the nitrogen and energy cycle of the entire system, and in this study, accounting for 33.1%, it was the most dominant bacterial phyla. *Actinobacteria* is the main nutrient supply in the soil. It exists in drier environments such as soil and air in the form of spores, and it is one of the most widespread bacteria in the ecosystem. The *Actinomycete* phyla accounted for 14.02% of the total number of bacterial

phyla and was one of the four most dominant bacterial phyla in the fields. *Chloroflexi* is a Gram-negative bacterium that can rely on light energy for photosynthesis to adapt to the environment under stress. The relative abundance of *Chloroflexi* in the cultivated fields was significantly lower ($p$-Value < 0.05). This finding was consistent with that of previous research [1,34,37,42]. This result indicates that continuous cropping decreased the relative abundance of plant-beneficial bacteria, which is important for crop production.

The UniFrac-weighted principal coordinate analysis (PCoA) showed evident variations in the bacterial community structure across the eight different continuous cropping fields and fallow field as control. In this study, LH was clustered together and separated from the other soil samples. The distance between them was relatively large, while the distance between different continuous cropping fields was relatively small. The results showed that continuous cropping significantly changed the rhizosphere bacterial community structure. These results were consistent with the findings of Xiong et al. [18], that long-term continuous cropping of black pepper showed noticeable variations in the bacterial community structure. Li et al. [46] found that the soil microbial community composition and structure among three peanut fields was altered significantly with different monoculture histories using 454-pyrosequencing analysis [47].

## 5. Conclusions

Our studies show that continuous cropping caused a significant change in soil physicochemical properties and enzyme activities, and these were different between cultivated and fallow fields. The continuous cropping of *C. chinensis*, maize, and other plant species significantly changed the bacterial community composition, structure, and diversity, and we found more diverse bacterial communities in the fallow field. Thus, our findings provide a foundation for future agricultural research to improve the microbial activity and increase crops/cash-crops output in continuous cropping fields, which is critical for *C. chinensis* production.

**Supplementary Materials:** The following are available online at https://www.mdpi.com/article/10.3390/agriculture11121224/s1, Figure S1: Composition of the bacterial community, Figure S2: Rarefaction curve of the bacterial community, Figure S3: Hierarchical clustering analysis, Table S1: Redundancy analysis of soil physicochemical properties and bacterial phyla, Table S2: Redundancy analysis of soil enzyme activities and bacterial phyla.

**Author Contributions:** Conceptualization, X.W.; Data curation, M.M.A., Z.G., T.Y., D.T., M.J.A. and X.W.; Formal analysis, M.M.A. and Q.P.; Funding acquisition, X.W.; Investigation, M.M.A., Q.P. and W.Y.; Methodology, M.M.A., Q.P., T.Y., O.Z. and X.W.; Project administration, X.W.; Resources, Q.P., Z.G., T.Y., D.T., O.Z., W.Y. and M.J.A.; Software, M.M.A., Z.G., D.T., O.Z., W.Y. and M.J.A.; Supervision, X.W.; Validation, M.M.A., T.Y. and X.W.; Visualization, M.M.A. and X.W.; Writing—original draft, M.M.A.; Writing—review & editing, M.M.A. and X.W. All authors have read and agreed to the published version of the manuscript.

**Funding:** This research was financially supported by the National Key R&D Program. Specialized Research on Modernization of Traditional Chinese Medicine. Large-scale cultivation of Huanglian (*Coptis chinensis* Franch) and two other Chinese medicinal plants with high values and their roles in the precise alleviation of poverty (2017YFC1701000).

**Institutional Review Board Statement:** Not applicable.

**Informed Consent Statement:** Not applicable.

**Data Availability Statement:** Data are now publicly available through the NCBI Sequence Read Archive (Accession# PRJNA783784 ID# 939928).

**Acknowledgments:** We thank Shanghai Majorbio Bio-Pharm Technology Co., Ltd., for providing MiSeq Sequencing and basic bioinformatics analyses software support. Thanks to the Chinese Scholarship Council (CSC) for providing scholarships for our studies.

**Conflicts of Interest:** The authors have no conflict of interest in this manuscript.

## Abbreviations

| | |
|---|---|
| PcoA | Principal Coordinate Analysis |
| NGS | Next-Generation Sequencing |
| RDA | Redundancy Analysis |
| OTUs | Operational Taxonomic Units |
| PICRUSt | Phylogenetic Investigation of Communities by Reconstruction of Unobserved States |
| KEGG | Kyoto Encyclopedia of Genes and Genomes |

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
