# Peer review of "Continuous Cropping Changes the Composition and Diversity of Bacterial Communities: A Meta-Analysis in Nine Different Fields with Different Plant Cultivation"

_agriculture, doi:10.3390/agriculture11121224_

Round 1

Reviewer 1 Report

Alami et al. studied the diversity of bacterial community in continuous cropping change fields. Authors also analyzed soil pH, the total content of nitrogen, phosphorus, and potassium, and soil enzyme activities in cropping change fields. Soil microbial communities are intricately linked to ecosystem functioning because they play important roles in carbon and nitrogen cycling. Therefore, there is a need to understand that how soil microbial communities will be affected by cropping system or cultural practices. In this paper, authors tried to answer these questions in reference to cropping systems. This study has significance importance to define microbial community stability and resistance against disturbance. I think this paper has interesting findings with valuable contribution

Some general comments

  • Change the title (present title represents a general statement; different crop plants are usually select their own microbial community as per their needs).
  • Also remove full stop (.) in the title.
  • Authors used different crop fields and their soil in this study. So, as per my suggestion, don’t focus on Chinese goldthread (Coptis chinensis ) in abstract and introduction part.
  • Add one paragraph in introduction part on need of this type of study (Usually, farmers use different cropping systems in their field, and crop plants select soil microbial community as per their need). What are you trying to suggest through this article? (mention about that in last paragraph of Introduction).
  • Statistical analysis part is missing in the main text as well as in the figure legends. Also add in table footnotes.
  • Rewrite conclusion part with giving suggestions for future studies, don’t write result here again. Talk about stability and resilience of soil bacterial community.   

Author Response

Reviewer 1

General Comments: Alami et al. studied the diversity of bacterial community in continuous cropping change fields. Authors also analyzed soil pH, the total content of nitrogen, phosphorus, and potassium, and soil enzyme activities in cropping change fields. Soil microbial communities are intricately linked to ecosystem functioning because they play important roles in carbon and nitrogen cycling. Therefore, there is a need to understand that how soil microbial communities will be affected by cropping system or cultural practices. In this paper, authors tried to answer these questions in reference to cropping systems. This study has significance importance to define microbial community stability and resistance against disturbance. I think this paper has interesting findings with valuable contribution.

Response: We appreciate the positive feedback from the reviewer. As suggested by the reviewer, we have reviewed the entire manuscript carefully and have removed redundancies, as shown in the revised manuscript.

Point 1: Change the title (present title represents a general statement; different crop plants are usually select their own microbial community as per their needs).

Also remove full stop (.) in the title.

Response 1: We thank the reviewer for this comment. We have changed the title and added some specific words to remove the general statement. And removed the full stop(.) from the title.

Point 2: Authors used different crop fields and their soil in this study. So, as per my suggestion, don’t focus on Chinese goldthread (Coptis chinensis) in abstract and introduction part.

Response 2: thanks for your comments. This study aims to find a suitable plant rotation for the C. chinensis and to increase the biological activities of the soil microbial community by selecting a suitable plant, thus, at the beginning of the abstract and introduction we focused on C. chinensis and its medicinal value.  This study helps the farmers and the production companies for selecting a better cultivation system and improve the quality of C. chinensis .

Point 3: Add one paragraph in introduction part on need of this type of study (Usually, farmers use different cropping systems in their field, and crop plants select soil microbial community as per their need). What are you trying to suggest through this article? (mention about that in last paragraph of Introduction).

Response 3: Thank you for your comments and suggestion. This paper is suggesting a better cultivation system and suitable plant rotation for the sustainable production of C. chinensis, we have revised the introduction and added the paper’s suggestion for the readers, researcher, and farmers in the last paragraph of the introduction. 

Point 4: Statistical analysis part is missing in the main text as well as in the figure legends. Also add in table footnotes.

Response 4:  Thanks for your comments.  We have added the statistical analysis method and materials in the method and materials section line 158 revised manuscript.

Point 5: Rewrite conclusion part with giving suggestions for future studies, don’t write result here again. Talk about stability and resilience of soil bacterial community

Response 5: thank you for your invaluable suggestion. We have revised the conclusion accordingly and added the study's suggestions.

Reviewer 2 Report

  • The presented goals are important and valuable. I have read this manuscript with interest.
  • The article concerns diversity and composition of the rhizospheric bacterial communities in 8 different continuously cropping fields with C. chinensis and fallow field.
  • The authors found significant differences in soil physicochemical properties (nutrient content - NPK and soil pH) and enzyme activity. The structure of bacterial communities has been significantly changed by the continuously cropping and plant species (crops). The most significant impact on the soil bacterial community was the continuously cropping of maize and P. multiflorum (the full name of this species is missing throughout the whole article).
  • I have to say it was a pleasant reading this paper, since it is detailed and sound, especially results chapter. The work is methodically well designed and the results are statistically well analyzed. However, in some places it needs to be improved and supplemented with information.
  • In the materials and methods chapter, in its particular subsections there should be reference to the literature where these methods are described. Nothing is known about the intensity of cultivation - agrotechnical treatments, fertilization, treatments to protect plants against diseases and pests. In my opinion they can be quite important to discuss, especially fertilization - mineral or natural, organic, all this affects the properties of the soil and the microbiological composition.
  • The results are rather well presented, illustrated and described. The discussion, on the other hand, could be fuller with more vocations. There are many studies showing changes in the physicochemical properties of soil under the influence of the cultivation of various crops depending on the cultivation system.
  • In addition, there are many different minor errors - such as capitalization, no spaces, incomplete species names, unclear captions and explanations of tables and figures. These remarks are marked in the manuscript and accompanied by comments.
  • In the bibliography in some items (articles) there is a missing page range. Item No.26 - Change the case of letters.
  • Also minor remarks in caption of tables and figures in the supplementary materials should be improved:
  • I suggest that a native speaker improves the language, since in some parts it could be slightly improved.
  • In my opinion, it would be better to give the meaning of the abbreviation immediately after its introduction, eg. LZ - continuous cropping of C. chinesis for 5 years; HY2 - maize ...etc.
  • Table S1 - analysis - maybe lowercase? and complete the title - properties of what? (soil is missing)
  • Table S2 - I don't understand why Urease and Sucrose are capital letter? No consistency - phosphatase is lower letter?

Author Response

Reviewer 2

General Comments:

  • The presented goals are important and valuable. I have read this manuscript with interest.
  • The article concerns diversity and composition of the rhizospheric bacterial communities in 8 different continuously cropping fields with C. chinensis and fallow field.
  • The authors found significant differences in soil physicochemical properties (nutrient content - NPK and soil pH) and enzyme activity. The structure of bacterial communities has been significantly changed by the continuously cropping and plant species (crops). The most significant impact on the soil bacterial community was the continuously cropping of maize and P. multiflorum (the full name of this species is missing throughout the whole article).
  • I have to say it was a pleasant reading this paper, since it is detailed and sound, especially results chapter. The work is methodically well designed and the results are statistically well analyzed. However, in some places it needs to be improved and supplemented with information.

Response: We sincerely thank the reviewer for constructive criticisms and valuable comments, which were of great help in revising the manuscript. Accordingly, the revised manuscript has been systematically improved with new information and additional interpretations. We have revised the figures and the entire manuscript accordingly. We have added the full name of P. multiflorum in the abstract.

Point 1: In the materials and methods chapter, in its particular subsections there should be reference to the literature where these methods are described. Nothing is known about the intensity of cultivation - agrotechnical treatments, fertilization, treatments to protect plants against diseases and pests. In my opinion they can be quite important to discuss, especially fertilization - mineral or natural, organic, all this affects the properties of the soil and the microbiological composition.

Response 1: Thank you! We found your comments extremely helpful. In this study, we have selected the fields which were under control and treated equally. Fertilizer was applied equally and no pesticide and insecticide apply in the fields for one year. For maize and P. multiflorum fields we have selected two fields with different continuous cropping times for 2 years and 6 years respectively.  The soil samples have been taken at one time at the end of the year before the harvesting. the condition was controlled for one year.  

Point 2: The results are rather well presented, illustrated and described. The discussion, on the other hand, could be fuller with more vocations. There are many studies showing changes in the physicochemical properties of soil under the influence of the cultivation of various crops depending on the cultivation system.

Response 2: Thanks for your comments.  The discussion part is revised and added some more information. The effect of continuous cropping on soil physicochemical properties and soil enzyme activities and soil microbial community has been reported by many researchers previously in some individual crops/plants but to compare different plants at the same time and condition is not reported before. therefore, we have decided to investigate the soil bacterial community composition and diversity in some very common crops/vegetables which are used as crop rotation for C. chinensis to find the relation between the plants and microbiome which was significant.

Point 3: In addition, there are many different minor errors - such as capitalization, no spaces, incomplete species names, unclear captions and explanations of tables and figures. These remarks are marked in the manuscript and accompanied by comments.

Response 3: We appreciate your comments and suggestions, we have revised the whole manuscript accordingly and removed all the errors and mistakes, however, we couldn’t find the highlighted or marks section in the manuscript that you have mentioned in the comment, it might be because of the technical problem.   However, I am looking forward to hearing from you if further revision is needed.

Point 4: In the bibliography in some items (articles) there is a missing page range. Item No.26 - Change the case of letters.

Response 4:  Thank you so much for catching these glaring and confusing errors. The bibliography is automatically generated by Mendeley and there were some mistakes and appropriate format. We have updated the bibliography and manually changed the references.   

Point 5: Also minor remarks in caption of tables and figures in the supplementary materials should be improved:

Response 5: Thank you for your comments. The supplementary figures and tables were checked carefully and remove all the misspellings and mistakes.  

Point 6: I suggest that a native speaker improves the language, since in some parts it could be slightly improved.

Response 6: Thank you for your comments.  The manuscript is revised by the corresponding and co-authors carefully and improved the English quality, for further improvement we will send for proofreading after peer review.

Point 7: In my opinion, it would be better to give the meaning of the abbreviation immediately after its introduction, eg. LZ - continuous cropping of C. chinesis for 5 years; HY2 - maize ...etc.

Response 7: Thank you for your assessment. We have added the meaning of the abbreviations immediately after their first appearance in the manuscript.

Point 8: Table S1 - analysis - maybe lowercase? and complete the title - properties of what? (soil is missing)

Response 8: Thank you for your comments. There was a misspelling and we have corrected the sentence.

Point 9: Table S2 - I don't understand why Urease and Sucrose are capital letter? No consistency - phosphatase is lower letter?

Response 9: Thank you for your observation. It was a misspelling and corrected. All are enzyme activities and are supposed to be in lower case.

Reviewer 3 Report

Manuscript Number: Agriculture-1462457

Title: Continuous cropping changes the composition and diversity of bacterial community: A meta-analysis in nine different fields.

Comments:

This manuscript investigates effect of different plant species and continuous cropping on soil bacterial community, which has not been studied before. Authors compared soil quality as well as soil bacterial community of different continuous cropping fields and fallow field without plantation. The results showed that the similarity of the bacterial community in the same crop rotation was higher. This study provides a better selection for the alleviation of continuous cropping obstacles of C. chinensisis. In general, the writing needs to be improved grammatically.

I suggest following questions for authors to consider: 

  1. The full name should be given when first appearance in text. e.g. Line 28: LZ and HS.
  2. The following sentences need revision to avoid misleading.

Line 67-68: “However, there 67 has been no research to compare different continuous cropping fields and plants and their 68 effect on soil physicochemical properties, enzyme activities, and bacterial community.”

Line 76-78: “Therefore, in this study, we have selected 8 different continuous cropping fields with different plants including C. chniensis continuous cropping fields and fallow field without plantation.”

Line438-439: “We have found that the soil organic matter content of LH field was the highest, which was higher than.”

  1. Avoid use abbreviation at the beginning of a sentence. e.g. Line 208, 226, 245, 426: LZ, HY2, HY6, HW2, HW6, HH, HB, HS, and LH; Line 252: V3-V4; Line 445: HY6.
  2. Discussion: The authors discussed correlation between plant species and SOM, soil nutrient availability.... Most of results are in agreement with previous studies. Are these studies conduced under same condition as current experiment? Discussion are suggested to develop.
  3. This study focused on 8 different continuous cropping fields and one fallow field after 2-6 years, were these field experimental field or local field? What’s the size of each field? Since the species planted were so different, were they fertilized the same rate every year? Did you take samples every year?
  4. In your abstract, you mentioned “This study provides a better selection for the alleviation of continuous cropping obstacles of C. chinensisis via various selections of rotational cropping patterns.”, how did the change of soil properties and microbial community in your experiment alleviate the continuous cropping obstacles of C. chinensisis ?

Once these changes are done, this paper could be of greater interest

Author Response

Reviewer 3

General comments:

This manuscript investigates effect of different plant species and continuous cropping on soil bacterial community, which has not been studied before. Authors compared soil quality as well as soil bacterial community of different continuous cropping fields and fallow field without plantation. The results showed that the similarity of the bacterial community in the same crop rotation was higher. This study provides a better selection for the alleviation of continuous cropping obstacles of C. chinensisis. In general, the writing needs to be improved grammatically.

I suggest following questions for authors to consider: 

Response: we sincerely appreciate and thank the reviewer for this constructive observation and valuable comments.

Point 1: The full name should be given when first appearance in text. e.g. Line 28: LZ and HS.

Response 1: thank you for your comments. We have added the full name of all abbreviations immediately after their first appearance in the manuscript.

Point 2: The following sentences need revision to avoid misleading.

Line 67-68: “However, there 67 has been no research to compare different continuous cropping fields and plants and their 68 effect on soil physicochemical properties, enzyme activities, and bacterial community.”

Line 76-78: “Therefore, in this study, we have selected 8 different continuous cropping fields with different plants including C. chniensis continuous cropping fields and fallow field without plantation.”

Line438-439: “We have found that the soil organic matter content of LH field was the highest, which was higher than.”

Response 2: we appreciate your invaluable suggestion. The sentences are revised in the manuscript and highlighted for better findings.

Point 3: Avoid use abbreviation at the beginning of a sentence. e.g. Line 208, 226, 245, 426: LZ, HY2, HY6, HW2, HW6, HH, HB, HS, and LH; Line 252: V3-V4; Line 445: HY6.

Response 3: thank you for your comments. We have revised the manuscript and removed or changed all the abbreviations from the beginning of the sentences.

Point 4: Discussion: The authors discussed correlation between plant species and SOM, soil nutrient availability.... Most of results are in agreement with previous studies. Are these studies conduced under same condition as current experiment? Discussion are suggested to develop.

Response 4:  we found your comments very useful and appreciate them. We have checked the discussion part and there was some misleading, the works which were cited in our paper is not similar to our work, they have found the effect of continuous cropping on the organic matter but did not compare different plants at the same time, we just mentioned that continuous cropping could the soil content of organic matter. the sentences are revised accordingly, for further revision we will be glad to hear from you.

Point 5: This study focused on 8 different continuous cropping fields and one fallow field after 2-6 years, were these field experimental field or local field? What’s the size of each field? Since the species planted were so different, were they fertilized the same rate every year? Did you take samples every year?

Response 5: We are so thankful for your invaluable comments. Regarding the experiment field, I have to say the fields were not the experimental fields we have selected some very common crops/vegetables local fields which are being used as crop rotation for C. chinensis in lichuan city.  The size of the fields was varied from plants to the plants in the same area. We have chosen 8 different fields and controlled the fertilizer, pesticide, insecticide, and organic fertilizer for one and treated the fields equally for one year, for the fallow fields we have selected the fields which were not cultivated for many years. The soil samples were taken once after one year of controlling before the harvesting.  

Point 5: In your abstract, you mentioned “This study provides a better selection for the alleviation of continuous cropping obstacles of C. chinensisis via various selections of rotational cropping patterns.”, how did the change of soil properties and microbial community in your experiment alleviate the continuous cropping obstacles of C. chinensisis?

Response 6: thank you for your comments. C. chinensis is an important medicinal plant and it’s the only source of income for the Lichuan people (Lichuan city Hubei province, China). And the farmers for improving the soil quality and fertility try to cultivate different crops and vegetables after harvesting the C. chinensis. Due to the limitation of land area in Lichuan city some crops/ vegetables are being cultivating successively for many years without crop rotation before the cultivation of C. chinensis, based on the local authority’s reports some crops/vegetables rotation increase the incidence of rot root and other diseases and its very harmful for the C. chinensis, therefore we have selected some very common crops/vegetable fields which were continuously cultivated for several years to find out the effect of plants and continuous cropping on soil bacterial community and compare them. As the microbial community is one of the most important factors for soil health and quality and directly influence the C. chinensis yield and quality and this is reported by many previous works, therefore, we have decided to find whether there is any relation between plant species and soil microbial community under a continuous cropping system or not and we have found a significant correlation between plant species and years of cultivation and soil bacterial community composition and diversity. This study provides a foundation for future agricultural researches to improve biological activity and increase crops/cash-crops productivity under a continuous cropping system, improve nutrient cycling in the soil, and mitigate the continuous cropping obstacles.
